# Cloning and Functional Study of *AmGDSL1* in *Agropyron mongolicum*

**DOI:** 10.3390/ijms25179467

**Published:** 2024-08-30

**Authors:** Xiuxiu Yan, Xiaojuan Wu, Fengcheng Sun, Hushuai Nie, Xiaohong Du, Xiaolei Li, Yongyu Fang, Yongqing Zhai, Yan Zhao, Bobo Fan, Yanhong Ma

**Affiliations:** 1Agricultural College, Inner Mongolia Agricultural University, Hohhot 010019, China; yanxx411@126.com (X.Y.); wuxj@emails.imau.edu.cn (X.W.); niehs@imau.edu.cn (H.N.); duxiaohong2024@126.com (X.D.); lixiaoleihlj@163.com (X.L.); zhaiyongqing66@126.com (Y.Z.); fanbobo19@126.com (B.F.); 2Inner Mongolia Academy of Agricultural & Animal Husbandry Sciences, Hohhot 010031, China; sfcnmnky@sina.com (F.S.); 1984fyy@163.com (Y.F.); 3College of Grassland, Resources and Environment, Inner Mongolia Agricultural University, Hohhot 010010, China; zhaoyannmg@sina.com

**Keywords:** *Agropyron mongolicum*, *AmGDSL1* gene, drought resistance, antioxidant activity

## Abstract

*Agropyron mongolicum* Keng is a diploid perennial grass of triticeae in gramineae. It has strong drought resistance and developed roots that can effectively fix the soil and prevent soil erosion. GDSL lipase or esterases/lipase has a variety of functions, mainly focusing on plant abiotic stress response. In this study, a *GDSL* gene from *A. mongolicum*, designated as *AmGDSL1*, was successfully cloned and isolated. The subcellular localization of the *AmGDSL1* gene (pCAMBIA1302-*AmGDSL1*-EGFP) results showed that the AmGDSL1 protein of *A. mongolicum* was only localized in the cytoplasm. When transferred into tobacco (*Nicotiana benthamiana*), the heterologous expression of *AmGDSL1* led to enhanced drought tolerance. Under drought stress, *AmGDSL1* overexpressing plants showed fewer wilting leaves, longer roots, and larger root surface area. These overexpression lines possessed higher superoxide dismutase (SOD), peroxidase (POD), catalase (CAT), and proline (PRO) activities. At the same time, the malondialdehyde (MDA) content was lower than that in wild-type (WT) tobacco. These findings shed light on the molecular mechanisms involved in the *GDSL* gene’s role in drought resistance, contributing to the discovery and utilization of drought-resistant genes in *A. mongolicum* for enhancing crop drought resistance.

## 1. Introduction

During agricultural development, the demand for water is high, and drought seriously affects agricultural production [1]. Generally, massive amounts of ROS (reactive oxygen species) are produced when plants are exposed to drought stresses [2]. Plants have developed intricate defense mechanisms to cope with excess ROS and protect cells from damage, which include superoxide dismutase (SOD), peroxidase (POD), and catalase (CAT) antioxidant enzyme systems [3]. In plants, a large number of drought-related genes have been identified and validated, such as the *MYB*, *NAC*, *WRKY* [4,5], and *bHLH* genes [6,7]. Three *TaMYB31* homeologous genes from hexaploid wheat (*Triticum aestivum* L.), were cloned and characterized. Ectopic expression of the *TaMYB31-B* gene in Arabidopsis (*Arabidopsis thaliana*) affected plants’ growth and enhanced drought tolerance [8]. *TaMYBsm1* genes from wheat play an important role in transgenic Arabidopsis drought stress tolerance through upregulation of *DREB2A*, *P5CS1,* and *RD29A* [9]. Wheat under drought stress was analyzed in a genome-wide association study, and it was found that the wheat *NAC* gene (*TaNAC071-A*) is closely associated with drought resistance. Knockdown of the *TaNAC071-A* gene reduces wheat drought resistance, whereas overexpression of *TaNAC071-A* enhances plant drought resistance [10]. Furthermore, overexpression of *TtNAC2A* in Arabidopsis plants showed enhanced drought resistance [11]. *TaWRKY1-2* silencing in wheat increases the MDA content, reduces the contents of proline and chlorophyll and the activities of antioxidant enzymes, and inhibits the expression levels of antioxidants (POD, CAT, and SOD) under drought stress [12]. *TabHLH27* knockout reduced wheat drought tolerance, yield, and water use efficiency [13]. 

Notably, *GDSL* genes are involved in various key physiological processes in plants, such as cuticle development, seed oil storage, male fertility, and biotic and abiotic stress responses [14,15]. The *GDSL* esterases/lipase (GELP) has been reported to be one of the largest hydrolase superfamilies. Its name is based on the N-terminal, which has a conserved GDSL motif (Gly-Asp-Ser-Leu) [16,17]. Under stress conditions such as drought and high salt, the expression level of *GDSL* lipase will undergo significant changes. Overexpression of *GhirGDSL26* in *Arabidopsis thaliana* resulted in longer root lengths, higher germination rates, and survival rates under drought resistance [18]. The function of the *CcGDSL* gene was analyzed through a cis-elements study. The expression of the *CcGDSL* gene is mainly influenced by fluctuating stressful situations, as well as by light and plant hormones. *CcGDSL21*, *CcGDSL63*, *CcGDSL64,* and *CcGDSL92* may be involved in regulating drought stress in pigeon pea (*Cajanus cajan* (L.) Druce) [19]. After salt stress treatment, weighted gene co-expression network analysis (WGCNA) showed that *CilGDSL41.11*, *CilGDSL39.49*, *CilGDSL34.85,* and *CilGDSL41.01* were significantly associated with salt stress in pecan leaves (*Carya illinoensis* K. Koch) [20]. *CaGLIP1* transgenic plants showed drought tolerance during seed germination and plant growth. Differential expression of *AtRD26A*, *AtADH, and AtRab18* genes was responsive in drought [21]. The expression of *CpGLIP1* was increased by treatments of drought, cold, jasmonic acid, and gibberellic acid. After drought and cold treatment, overexpression of *CpGLIP1* in Arabidopsis thaliana and poplar (*Populus tremula* × *Populus alba*) plants resulted in higher rates of growth and survival and better physiological indices than in the wild-type (WT) plants. Overexpression of *CpGLIP1* can improve tolerance of drought and cold [22]. The *GDLS* genes will be useful as targets for genetic modification in improving crop yield, quality, or both.

*A. mongolicum* Keng diploid (2n = 14) is a perennial herbaceous plant of the triticeae tribe in the poaceae family. As a high-quality forage, it has characteristics such as drought resistance, cold resistance, and wind and sand resistance. *A. mongolicum* Keng is also considered a primitive close relative of some crops, which can provide an abundance of drought stress-responsive genes [23]. An abundance of drought stress-responsive genes in *A. mongolicum* Ken were excavated by sequencing technology. In *A. mongolicum* during drought stress, there were 2171 unigene differences of expression [24]. The gene *AmbHLH148* from *A. mongolicum* was cloned and isolated; the overexpression of *AmbHLH148* in transgenic tobacco acted as a positive regulator under drought stress by enhancing the plant’s antioxidant capacity [25]. An *AP2/EREBP* transcription factor gene, cDNA from *Agropyron mongolicum* Keng, was isolated through RT-PCR and RACE techniques and named *MwAP2/EREBP*. *MWAP2/EREBP* has tissue and temporal specificity, which suggested that the *MwAP2/EREBP* gene might be involved in the drought-resistant physiological process of *A. mongolicum* Keng [26]. In addition, some miRNAs have also been identified as being involved in the drought stress response in *A. mongolicum* Keng. Under drought conditions, a total of 114 miRNAs were identified in *A. mongolicum*: amo-miR21, amo-miR5, and amo-miR62 were participants in drought stress response and genetically stable [27]. Fan et al. identified the key miRNA-target gene under drought stress by using multi omics; osa-miR444a-3p.2-*MADS47*, bdi-miR408-5p_1ss19TA-*CCX1*, tae-miR9774_L-2R-11ss11GT-*CARC*, ata-miR169a-3p-*PAO2*, and bdi-miR528-p3_2ss15TG20CA-*HOX24* were discovered to play central roles in the regulation of drought-responsive genes [28]. A large number of drought-resistance genes were excavated in *A. mongolicum* Keng. In previous studies based on transcriptome sequencing data obtained from *A. mongolicum* in drought stress (NCBI, PRJNA742257) [29], 52 *GDSL* genes were identified through bioinformatics screening. Physical and chemical properties analysis, phylogenetic tree analysis, and qRT-PCR analysis were performed on 52 *GDSL* lipase genes. *AmGDSL1* (DN29240_c0_g3) exhibited significant upregulation in response to drought stress. To understand its potential role in drought tolerance, *AmGDSL1* was heterologously inserted into tobacco plants. This research offers valuable insights into the function of *AmGDSL1* in drought tolerance and offers genetic resources that could enhance drought resistance in other crops.

## 2. Results

### 2.1. Analysis of AmGDSL1 Gene Expression in A. mongolicum

Under normal growth conditions, the *AmGDSL1* gene is expressed in the roots, stems, and leaves of *A. mongolicum* (Figure 1A). The *AmGDSL1* gene had the highest expression level and showed significant differences in leaves. The expression level of *AmGDSL1* gene in leaves was 2.7 times higher than that in stems and is 1.9 times higher than that in roots. The expression level of the *AmGDSL1* gene in the stem was 1.4 times higher than that in the root.

At the same time, the seedlings of *A. mongolicum* were subjected to a simulated osmotic stress treatment with 25% PEG-6000, and the expression level of the *AmGDSL1* gene was significantly different in dehydration stress periods. There was a significant difference between the expression level of the *AmGDSL1* gene in the *A. mongolicum* control (CK) and during the different osmotic stress periods (Figure 1B). At different osmotic stress periods, the expression level of the *AmGDSL1* gene was significantly different. Notably, as the osmotic stress duration increased, the expression level of the *AmGDSL1* gene revealed an overall increasing trend. At 5 days, the gene expression reached its highest level, approximately 6.6 times that of the control sample. The results showed that during the simulated osmotic stress treatment of *A. mongolicum* with 25% PEG-6000, the AmGDSL1 gene was upregulated in response to drought.

### 2.2. Cloning of AmGDSL1 Gene in A. mongolicum and Overexpression Vector Construction

The specific band obtained after PCR amplification using cDNA as a template is shown in Figure 2A. The band size was 634 bp, which is consistent with the size of the *AmGDSL1* gene. The sequencing results of the *AmGDSL1* gene and transcription sequence alignment showed a matching degree of 96.44% and proved that the *AmGDSL1* gene was successfully cloned (Figure 2C). The *AmGDSL1* gene and pCAMBIA1302-EGFP expression vector were cleaved using *NcoI* and *SpeI*. Simultaneously, the T4 ligase was used to ligase cleavage products. The ligase cleavage products amplified a specific band of about 634 bp, which preliminarily confirmed that the plant overexpression vector pCAMBIA1302-*AmGDSL1* was successfully constructed (Figure 2B).

### 2.3. Subcellular Localization of AmGDSL1 Gene

To verify the subcellular localization of the *AmGDSL1* gene, an expression vector, pCAMBIA1302-*AmGDSL1*-EGFP, was constructed, which had a 35S promoter and fused with green fluorescent protein (EGFP). Subsequently, this recombinant vector was injected into onion epidermal cells. Observations made through confocal microscopy revealed that the GFP signals originating from the control pCAMBIA1302-EGFP empty vector were distributed throughout the cell in the cell membrane, nucleus, and cytoplasm of the onion epidermal cells. In contrast, PCAMBIA1302-*AmGDSL1*-EGFP is mainly distributed in the cytoplasm. After treatment with sucrose solution, the onion cytoplasmic wall was separated. Clearly, pCAMBIA1302-*AmGDSL1*-EGFP fusion protein was exclusively localized in the cytoplasm (Figure 3).

### 2.4. Production and Identification of AmGDSL1 Transgenic Tobacco

The Agrobacterium liquid (LBA4404) containing the pCAMBIA1302-*AmGDSL1* recombinant plasmid was used to transform the sterile young tobacco leaves via the leaf disc method. Afterwards, co-culture, differentiation culture, and rooting culture were carried out to obtain regenerated tobacco plants. Using pCAMBIA1302-*AmGDSL1* plasmid as a positive control (P), wild-type tobacco as negative control (WT), and ddH_2_O as a blank control (CK), PCR amplification and agarose gel electrophoresis were performed. The results showed that the regenerated plants obtained specific amplified bands around 634 bp, with no bands in WT and CK, indicating that the *AmGDSL1* gene had been integrated into the tobacco genome (Figure 4A). qRT-PCR detection was performed on the leaves of transgenic tobacco-positive plants, and the results showed that among the 15 transgenic positive strains, the OE5 and OE6 strains had the highest expression level of the *AmGDSL1* gene (Figure 4B). 

### 2.5. Phenotype Observation of Transgenic Tobacco under Drought Stress

To verify the drought resistance of transgenic tobacco, *AmGDSL1* overexpression and wild-type tobacco (WT) plants were subjected to natural drought treatment by discontinuing watering. Under normal growth conditions, there is a significant difference between phenotypes in *AmGDSL1*-OE and WT plants. It was found that *AmGDSL1* overexpression tobacco plants have larger leaves and higher stems. Under drought conditions of 7 days, WT plants exhibited a high number of wilted leaves overall. In contrast, *AmGDSL1* overexpression tobacco showed a remarkable reduction in wilted leaves, and they were found at the bottom (Figure 5A). Simultaneously, the transgenic tobacco exhibited longer roots and a greater root surface area compared to the WT plants. This indicates that *AmGDSL1* overexpression can improve tobacco drought tolerance and enable a higher survival rate under drought stress by promoting growth of more roots (Figure 5B,C).

### 2.6. Overexpression of AmGDSL1 Enhanced Antioxidant Capacity of Tobacco

In order to verify the function of the *AmGDSL1* gene in the antioxidant defense mechanism, SOD, POD, CAT, PRO, and MDA measurements were performed on drought-treated *AmGDSL1* overexpression tobacco. In the WT and *AmGDSL1* transgenic lines, the trend of SOD, POD, and CAT activity was slightly higher than that in WT plants under normal conditions. Apparently, the *AmGDSL1* overexpression significantly increases the SOD, POD, CAT activity in drought stress. As shown in Figure 6, activity of SOD was induced by 1.32-fold in OE5 and 1.41-fold in OE6, activity of POD was induced by 1.13-fold in OE5 and 1.26-fold in OE6, and activity of CAT was induced by 1.42-fold in OE5 and 1.26-fold in OE6. In the WT and *AmGDSL1* transgenic lines, the trend of PRO content was consistent under normal conditions. After 7 days of drought stress, the content of PRO in *AmGDSL1* transgenic lines significantly increased, indicating that transgenic plants have stronger osmotic regulation levels under drought stress conditions. The content of PRO was induced by 1.27-fold in OE5 and 1.17-fold in OE6. In the WT and *AmGDSL1* transgenic lines, the trend of MDA content was slightly lower than that in WT plants under normal conditions. Conversely, the content of MDA in WT tobacco was significantly higher than that in *AmGDSL1* transgenic lines in drought stress. The content of MDA in WT tobacco was 1.51 times that in OE5 and 1.22 times that in OE6. Obviously, under drought stress conditions, the damage to the cell membrane of transgenic plants is relatively mild, while the damage to the membrane system of WT plants is severe.

### 2.7. Expression Regulation in AmGDSL1 Transgenic Tobacco

To further explore the function of *AmGDSL1* gene, the expression patterns of stress-responsive genes were detected in overexpression lines and WT plants using qRT-PCR (Figure 6F). Under normal conditions, the expression level of overexpressed tobacco’s *AmGDSL1* gene is significantly higher than that of wild-type tobacco. In drought stress, the expression levels of *AmGDSL1* genes were significantly induced in overexpression lines compared to that in WT plants. Correlation analysis showed that under drought stress, the SOD, POD, CAT, and MDA activity and proline content were positively correlated with gene expression (Figure 7). Apparently, the *AmGDSL1* in transgenic tobacco has a positive regulatory effect on drought stress.

## 3. Discussion

With the development of plant genome sequencing and bioinformatics, GDSL lipases have been discovered in various plants. There were 108 GDSL lipase proteins identified in Arabidopsis and 114 GDSL lipase proteins identified in *Oryza sativa* L [28]. There were 53, 65, and 96 *GDSL* genes identified in maize (*Zea mays* L.), watermelon (*Citrus lanatus*), and grape (*Vitis vinifera* L.) [30,31]. In addition, 597 *GELP* genes were identified from six Rosaceae genomes (i.e., *Fragaria vesca*, *Prunus persica*, *Prunus avium*, *Prunus mume*, *Pyrus bretschneideri*, and *Malus domestica*), based on phylogenetic tree analysis, and all *GELP* genes were further divided into ten subfamilies [31]. The diversity of these genes was revealed in terms of gene structure, evolutionary relationships, and expression patterns [32]. In previous studies, differentially expressed *GDSL* genes were identified in the transcriptome data of drought-treated *A. mongolicum* [18]. In this study, the *AmGDSL1* gene was identified in *A. mongolicum* under drought treatment, which may be associated with drought stress.

The expression of the *GDSL* gene shows diversity in different tissues and organs; for instance, *BrEXL6* is only highly expressed in buds and has lower expression levels in other tissues [33]. Lee et al. found that *GLIP1* is expressed at different developmental stages, such as in *Arabidopsis thaliana* seedling, flower, leaf, stem, and root, while *GLIP2* is only highly expressed at the root, stem, and seedling stages [17]. Twelve *GDSL* genes were analyzed by quantitative real-time PCR, and the results showed the expression levels of the 12 genes were different in the russet exocarp and green exocarp of pear (*Pyrus* spp.) at different fruit development stages [34]. Our study shows that the *AmGDSL1* gene has the highest expression level in the *A. mongolicum* leaves, followed by the roots, and the lowest expression level in the stems. To identify whether the *AmGDSL1* gene is induced by drought stress, a 25% PEG-6000-simulated drought stress treatment was performed on *A. mongolicum*. The results showed that except for on the fs1 day, the *AmGDSL1* expression levels were higher than in CK, showing an upregulated expression with prolonged drought time. Obviously, the *AmGDSL1* gene participates in plant drought stress.

The *GDSL* gene is widely distributed in plants and is involved in various abiotic stresses, antiviral activities, in seed germination, and biological metabolism. However, so far, only a few *GDSL* lipase genes have been cloned and validated in plants. Su et al. analyzed the expression patterns of soybean (*Glycine max*) *GmGELP* genes under different tissues and stress conditions through transcriptome sequencing. Seven stress-responsive *GmGELP* genes were screened and validated by qRT-PCR. Among the seven *GmGELP* genes, *GmGELP28* showed a significant response to drought, salt, and ABA treatments, and exhibited drought and salt tolerance characteristics in transgenic Arabidopsis and soybean plants. Therefore, *GmGELP28* was preliminarily identified as a potential candidate gene for improving soybean drought and salt tolerance [35]. Simone et al. found that *LOC-Os01g11760.1*, *LOC-Os07g44780.1*, and *LOC-Os10g25400.1* all belong to the GDSL lipase family and respond to drought stress in *Oryza sativa*. L roots and leaves [36]. Xiao et al. systematically studied the expression patterns of 19 *GDSL* lipases (*GGLs*) in Arabidopsis at different developmental stages and under hormone and abiotic stress treatments. They found that *GGL7*, *GGL14*, and *GGL26* responded to hormone and abiotic stress. At the same time, under severe drought stress, the drought tolerance of *GGL7, GGL14*, and *GGL26* mutants was significantly stronger than that of *GGL14-GGL26*, indicating that *GGL7*, *GGL14*, and GGL26 play a unique role in water retention [37]. Tang et al. identified the GDSL lipase gene (*OSP1*) in Arabidopsis thaliana. In drought stress, *OSP1* mutation resulted in significant reduction in leaf wax synthesis and occlusion of stomata, leading to increased epidermal permeability, decreased transpiration rate, and enhanced drought tolerance [38]. Liu et al. found overexpression of the *GhirGDSL26* gene in Arabidopsis thaliana resulted in enhanced drought resistance under drought stress. Silencing of *GhirGDSL26* resulted in a susceptible phenotype, higher MDA and H_2_O_2_ contents, and lower SOD activity and proline content [18]. The results demonstrated that *GhirGDSL26* plays a critical role in cotton drought stress tolerance. In our study, under drought stress treatment, compared to the WT, *AmGDSL1*-OE plant lines had fewer wilted leaves, which had longer roots and larger root surface area. Furthermore, *AmGDSL1*-OE plants had higher SOD, POD, CAT, and PRO activities than WT plants and lower MDA content than WT. Notably, a typical *GDSL* gene, *AmGDSL1,* was cloned from *A. mongolicum* and functionally verified in overexpressing tobacco. *AmGDSL1* overexpression lines showed significantly increased resistance to dehydration stresses. In our study, the drought tolerance function of the *AmGDSL1* gene was preliminary identified but its metabolic pathways and molecular mechanisms involved in dehydration stress response needed further research.

## 4. Materials and Methods

### 4.1. Plant Materials and Sample Preparation

The seeds of *A. mongolicum* were collected in the germplasm resource garden of the Inner Mongolia Agricultural University Grass Experiment Station. *A. mongolicum* seeds were selected and the palea and lemma were removed, rinsed with running water for 3 h and with 75% ethanol for 30 s, then rinsed with sterilized distilled water 5 times, disinfected with 4.3% NaClO for 4 min, and rinsed with sterilized distilled water 5 times. The seeds were placed in a germination box at 24 °C with 16 h/8h light and darkness in an artificial climate box (Ningbo, Zhejiang, China). When seeds germinated and grew about 8–10 cm, they were transplanted to a new germination box containing 1/5 Hoagland’s nutrient solution. When the *A. mongolicum* reached the three-leaf one-center stage of growth, 1/5 Hoagland’s nutrient solution containing 0.042 mol/L PEG-6000 (25% PEG-6000) was used to simulate osmotic treatment. Leaves were collected after drought for 0 days (CK), 1 day, 2 days, 3 days, 5 days, 7 days, and fs1 days (fs1 d indicates the sample collected after 24 h of restoration in 1/5 Hoagland’s nutrient solution). Three leaves from the same plant were used as three biological replicates. All samples were immediately frozen in liquid nitrogen and stored at −80 °C for RNA extraction.

The seeds of *Nicotiana benthamiana* are preserved in the laboratory (Agricultural College, Inner Mongolia Agricultural University). The seeds of *Nicotiana benthamiana* were vernalized at 4 °C for 2–3 days. They were then disinfected with 75% ethanol for 30 s, rinsed with sterile distilled water 5 times, and 4.3% NaClO for 4 min. The seeds were placed in MS solid culture medium, and after germination, they were transferred to a new MS solid culture medium for genetic transformation experiments. The wild-type tobacco *Nicotiana benthamiana* (WT) and AmGDSL1-OE tobacco were cultured to consistent growth at approximately 6 weeks of age, then subjected to drought treatment by cessation of watering for 7 days. Tobacco plants under drought treatment were observed to identify the phenotype characteristics, take indicator measurements, and extract RNA.

### 4.2. Cloning of AmGDSL1 Gene and Vector Construction

Total RNA was extracted from the *A. mongolicum* using TRNzol (Tiangen, Beijing, China). The total RNA was used to synthesize the cDNA of the genes using the Fast Quant cDNA Kit KR106 (Tiangen, Beijing, China). The ORF of *AmGDSL1* was amplified by RT-PCR, and the primer sequence (*AmGDSL1*-c-F/R) list in Appendix A. The purified PCR products were subcloned in the pEASY^®^-T1 Simple Cloning Vector, and the constructed vector was then transformed into Escherichia Trans1-T1 competent cells (TransGen, Beijing, China). After sequencing, the recombinant plasmid was named pEASY-*AmGDSL1*. pCAMBIA1302-EGFP plasmids (Abiowell, Changsha, China) and pEASY-*AmGDSL1* plasmids were extracted according to the plasmid extraction kit (TIANGEN DP103, Beijing, China) and were digested using *Nco*I and *Spe*I restriction endonucleases (Takara, Beijing, China). The pCAMBIA1302-*AmGDSL1*-EGFP recombinant vector was constructed by ligating the *AmGDSL1* product without a termination codon into the pCAMBIA1302-EGFP vector fragment. The enzyme digestion system included 0.5 μL *Nco*I, 0.5 μL *Spe*I, 2 μL 2 × K buffer, 10 μL pCAMBIA1302-EGFP/pEASY-*AmGDSL1,* and 7 μL ddH_2_O. The digestion process was carried out at 37 °C for 16 h. The enzyme digestion products used were DNA Gel/PCR Purification Miniprep Kit (Beiwo, Hangzhou, China). The ligation system consisted of a 2.5 µL of linearized pCAMBIA1302 vector, 1.5 µL AmGDSL1, 2 µL T4 DNA buffer, 1 µL DNA Ligation Kit (TaKaRa, Beijing, China), and 13 µL ddH_2_O. The digestion process was carried out overnight at 16 °C. The ligation was carried out using an ice for 25 min, a water bath at 42 °C for 45 s, and ice for 2 min, which transformed the ligation products into E. coli receptor TOP10 cells. Positive transformants were selected and placed on LB agar plates containing 50 µg/mL kanamycin for PCR analysis. After the PCR detection (AmGDSL1-F/R) of the recombinant plasmid, it was named pCAMBIA1302-*AmGDSL1*. The pCAMBIA1302-*AmGDSL1* plasmid was placed in liquid nitrogen for 5 min, then in a water bath at 37 °C for 5 min, and on ice for 5 min, and transformed into agrobacterium LBA4404 competent cells (Weidi, Shanghai, China). Positive transformants were selected on LB agar plates containing 50 µg/mL kanamycin for PCR analysis.

### 4.3. Subcellular Localization of AmGDSL1

The gene *AmGDSL1* was inserted into the pCAMBIA1302 vector, which was started by the CaMV-35S promoter and contained the EGFP fragment. The Agrobacterium tumefaciens with pCAMBIA1302-*AmGDSL1*-EGFP vector was activated and resurrected by 1/2MS liquid culture medium. The resuspension was adjusted to OD600:0.6 and the suspension was left at room temperature for 2 h, then used to infect onion epidermal cells. Simultaneously, the empty vector pCAMBIA1302-EGFP was used as a control. We spread the onion epidermis on MS medium and pre-treated it in the dark at 25 °C for 24 h. Then, the pre-treated onion epidermis was soaked in the resuspended Agrobacterium tumefaciens liquid for 20 min. Subsequently, the onion epidermal Agrobacterium liquid was dried using filter paper, and the side with the wound was spread flat on MS solid medium. The medium was incubated in a light incubator at a light cycle of 16/8 h and 25 °C for 48 h. The confocal laser microscope was used to observe the fluorescence signal of the fusion protein. The onion epidermal tissue was treated with 300 g/L sucrose solution for 5 min until the separation of the cell wall was observed.

### 4.4. Obtaining and Identifying of AmGDSL1 Transgenic Tobacco Plants

The pCAMBIA1302 plasmid containing the *AmGDSL1* gene was transformed into tobacco leaves by the Agrobacterium-mediated method (Figure 1). The immature leaves of the *Nicotiana benthamiana* were cut into 0.5 × 0.5 cm^2^ samples and placed in resuspended Agrobacterium solution for 8 min. The Agrobacterium liquid on the leaves was dried using filter paper, and leaves were transferred to a co-culture medium (MS+1 mg/L 6-BA) then incubated at 28 °C in the dark for 2–3 days. The co-cultured tobacco leaves was transferred to a differentiation medium (MS + 1 mg/L 6-BA + 300 mg/L TMT + 50 mg/L Kan) for bud differentiation. Subsequently, the differentiated buds were transferred to rooting medium (MS + 0.5 mg/L IAA + 300 mg/L TMT + 50 mg/L Kan) to induce rooting and enable the formation of small plants (Figure 8). DNA was extracted from regenerated plant leaves using a FastPure Plant DNA Isolation Mini Kit-BOX2 (Vazyme, Nanjing, China). The primer *AmGDSL1*-F/R was used for PCR detection to determine if the plants were positive.

### 4.5. Drought Stress Tolerance Assays of AmGDSL1 Transgenic Tobacco Plant

The wild-type tobacco (WT) and *AmGDSL1*-OE tobacco were cultured to consistent growth at approximately 6 weeks of age, then subjected to drought treatment by cessation of watering for 7 days. To investigate the drought resistance function of *AmGDSL1* gene in tobacco, the phenotypes of WT and *AmGDSL1*-OE plant lines were observed after 7 days of drought stress treatment. Three leaves from the same plant were used as three biological replicates. The activities of SOD (A001-3), POD (A084-3), CAT (A007-1-1), MDA (A003-1), and PRO (A107-1-1) in wild-type (WT) and *AmGDSL1*-OE plant lines at 0 d and 7 d of drought were evaluated using specific test kits (Nanjing Jiancheng Bioengineering Research Institute, Nanjing, China).

### 4.6. Analysis of AmGDSL1 Gene Expression

*A. mongolicum* leaves were collected after stress for 0 days (CK), 1 day, 2 days, 3 days, 5 days, 7 days, and fs1 days (fs1 d indicates the sample collected after 24 h of restoration in 1/5 Hoagland’s nutrient solution) for the *AmGDSL1* gene expression under osmotic stress. *A. mongolicum* roots, stems, and leaves were collected to analyze the *AmGDSL1* gene expression in different tissues and organs. WT and *AmGDSL1*-OE plant leaves were collected after drought stress at 0 days and 7 days for the *AmGDSL1* gene expression in WT and transgenic tobacco. All samples from the same plant were used as three biological replicates. The total RNA was used to synthesize the cDNA of the genes using the FastQuant cDNA Kit KR106 (Tiangen, Beijing, China). The qRT-PCR analysis was conducted using an MonAmpTM SYBR^®^ Green qPCR Mix (Monad, Suzhou, China) and the qRT-PCR primer sequence (*AmGDSL1*-q-F/R) list in Appendix A. The *U6* gene was used as an internal reference to accurately normalize each reaction for the genes of the *A. mongolicum*, respectively. The relative levels of the *AmGDSL1* gene in *A. mongolicum* were calculated using the 2^−∆∆CT^ [39] method. The *AmGDSL1* gene expression in different organizations of *A. mongolicum* and transgenic tobacco were calculated using the 2^−∆∆CT^ [39] method. 

### 4.7. Statistical Analysis

Data were collated and bar charts were drawn using Excel 2010 (Microsoft, Albuquerque, NM, USA). Data were statistically analyzed by one-way variance analysis by Duncan multiple range tests in SPSS 19.0 (IBM Inc, San Francisco, CA, USA) software. Differences were regarded as significant at *p* < 0.05. The Pearson correlation coefficients (r) were computed using R software (psych 2.2.9).

## 5. Conclusions

In this study, a typical *GDSL* gene *AmGDSL1* was cloned from *A. mongolicum* and functionally verified in overexpressing tobacco. *AmGDSL1* is a cytoplasm-localized protein. We also examined the increase in *AmGDSL1* gene expression induced by PEG6000 treatment of *A. mongolicum*. *AmGDSL1* overexpression lines showed significantly increased resistance to dehydration stresses. This study explores the *GDSL* gene related to dehydration stresses in *A. mongolicum*, providing reference for crop genetic improvement. Future research may improve crop resistance in gene breeding.

## Figures and Tables

**Figure 1 ijms-25-09467-f001:**
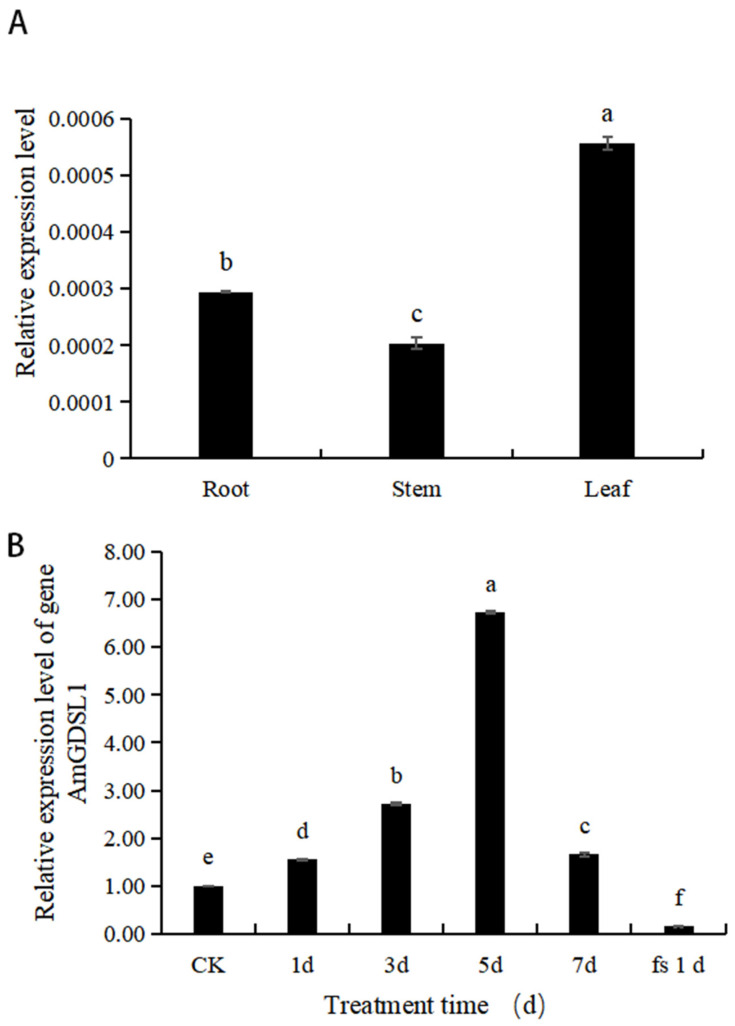
Expression of the *AmGDSL1* gene. (**A**) Expression level of the *AmGDSL1* gene in roots, stems, and leaves of *A. mongolicum*. (**B**) Relative expression of the *AmGDSL1* gene in *A. mongolicum* under drought stress. Different lowercase letters indicate significant differences at the 0.05 level (Duncan test).

**Figure 2 ijms-25-09467-f002:**
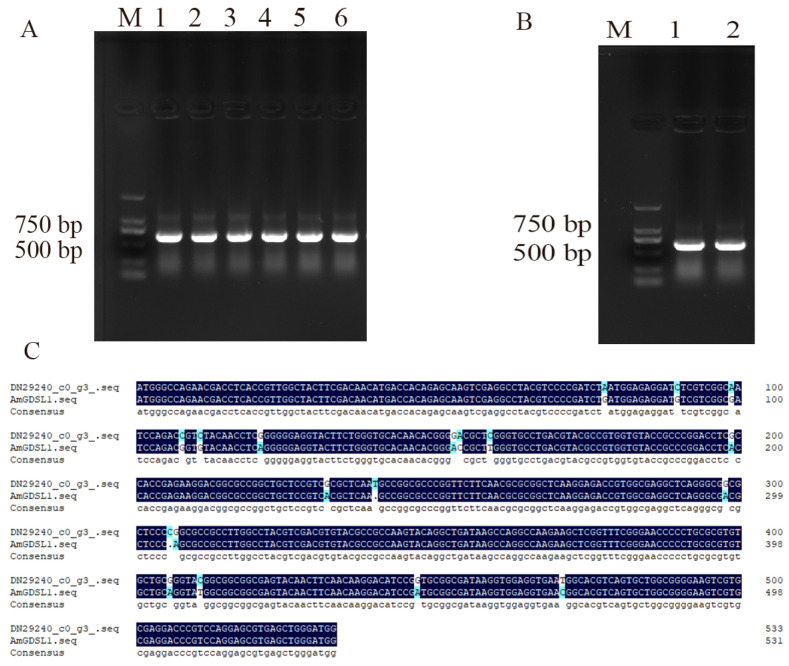
Cloning of *AmGDSL1* gene and vector construction. (**A**) PCR amplification electrophoresis detection of *AmGDSL1*. M:D2000 DNA Maker, 1–6: *AmGDSL1*. (**B**) PCAMBIA1302-*AmGDSL1* transformed Agrobacterium LBA4404 colony PCR electrophoretogram.M:D2000 DNA Maker, 1–6: *AmGDSL1*. (**C**) Alignment of *AmGDSL1* transcribed sequence with sequencing sequence. Black: Consistent sequence alignment. Blue: Base mutation.

**Figure 3 ijms-25-09467-f003:**
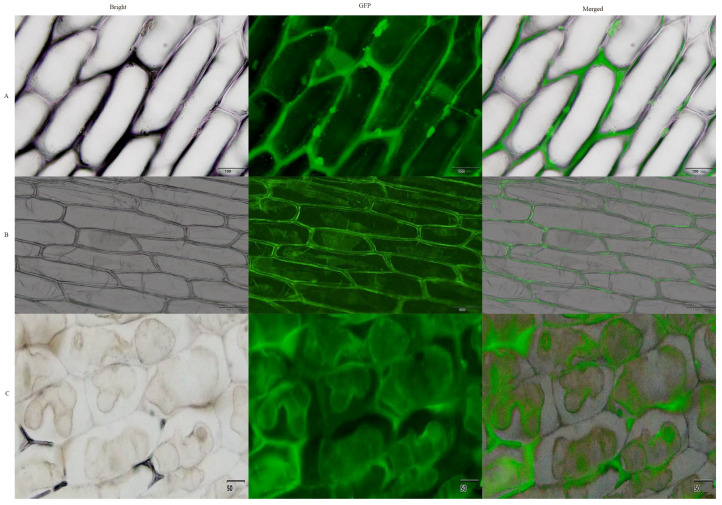
Subcellular localization of *AmGDSL1*. (**A**) EGFP positive control (pCAMBIA1302-EGFP). (**B**) pCAMBIA1302-*AmGDSL1* EGFP is independently expressed in onion epidermal cells. (**C**) Expression of pCAMBIA1302-*AmGDSL1*-EGFP after plasma wall separation. Scale bar = 50 μm.

**Figure 4 ijms-25-09467-f004:**
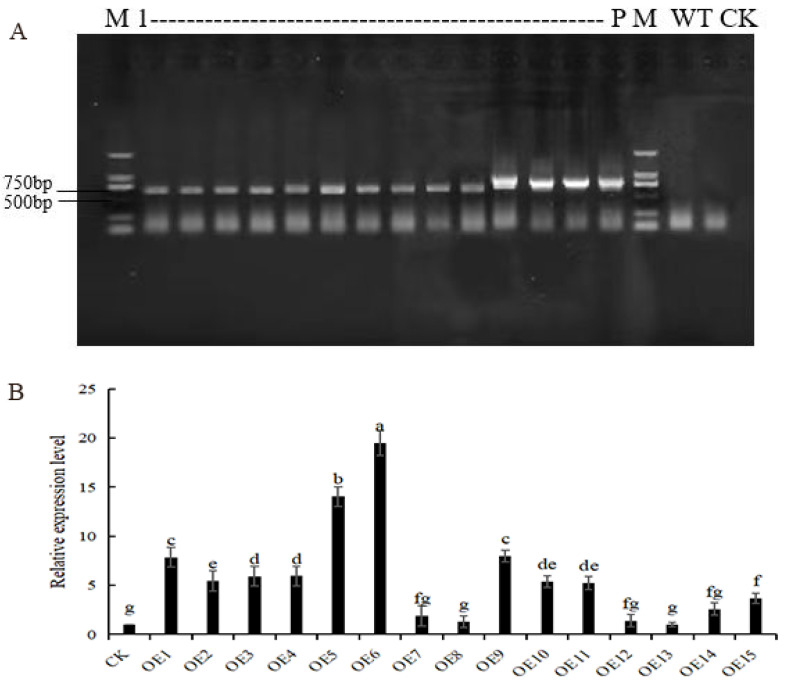
Obtaining and identifying of *AmGDSL1* transgenic tobacco plants. (**A**) Electrophoretogram of PCR assay of positive plants of *AmGDSL1* transgenic tobacco. (**B**) Relative expression levels of the *AmGDSL1* gene in transgenic plants. Different lowercase letters indicate significant differences at the 0.05 level (Duncan test).

**Figure 5 ijms-25-09467-f005:**
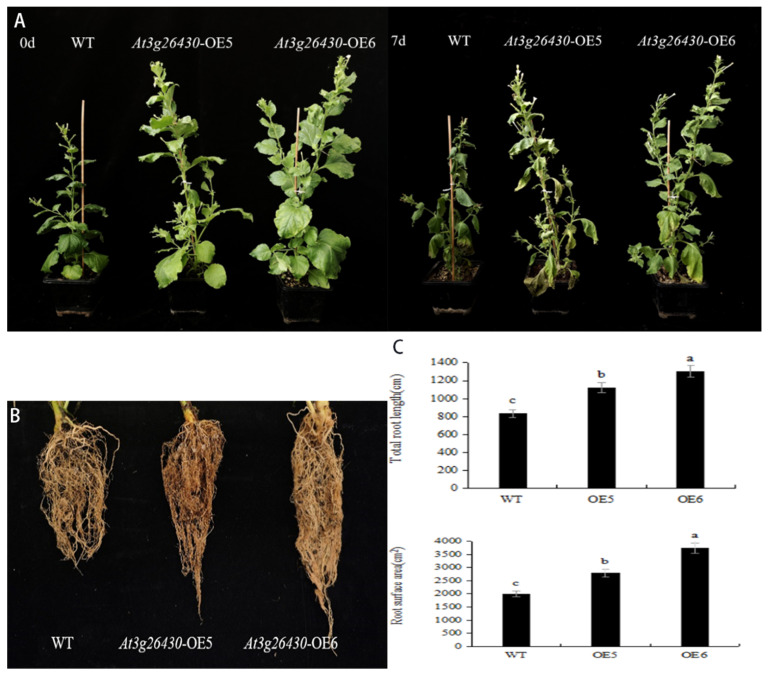
Drought resistance of *AmGDSL1*-transformed tobacco. (**A**) Phenotypic characteristics of transgenic plants under drought stress. (**B**) Root phenotype of transgenic. Root phenotypes of wild-type and transgenic tobacco under drought for 7 days. (**C**) Root system analysis of transgenic plants. Total root length of WT and transgenic tobacco under drought conditions for 7 days. Root surface area of WT and transgenic tobacco under drought conditions for 7 days. Different lowercase letters indicate significant differences at the 0.05 level (Duncan test).

**Figure 6 ijms-25-09467-f006:**
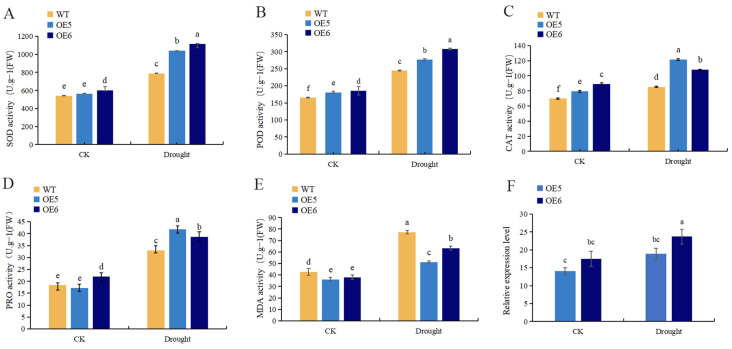
The antioxidant capacity and the AmGDSL1 expression in transgenic plants. (**A**) SOD activity of WT and transgenic plants. (**B**) POD activity of WT and transgenic plants. (**C**) CAT activity of WT and transgenic plants. (**D**) PRO content of WT and transgenic plants. (**E**) MDA content of WT and transgenic plants. Lowercase letters in the above graph represent significant differences at the 0.05 level. All data are expressed as the mean + SD. (**F**) The expression levels of the AmGDSL1 gene and in AmGDSL1-OE plants. Data are mean ± standard error of three replicates. Lowercase letters in the above graph represent significant differences at the 0.05 level (Duncan test).

**Figure 7 ijms-25-09467-f007:**
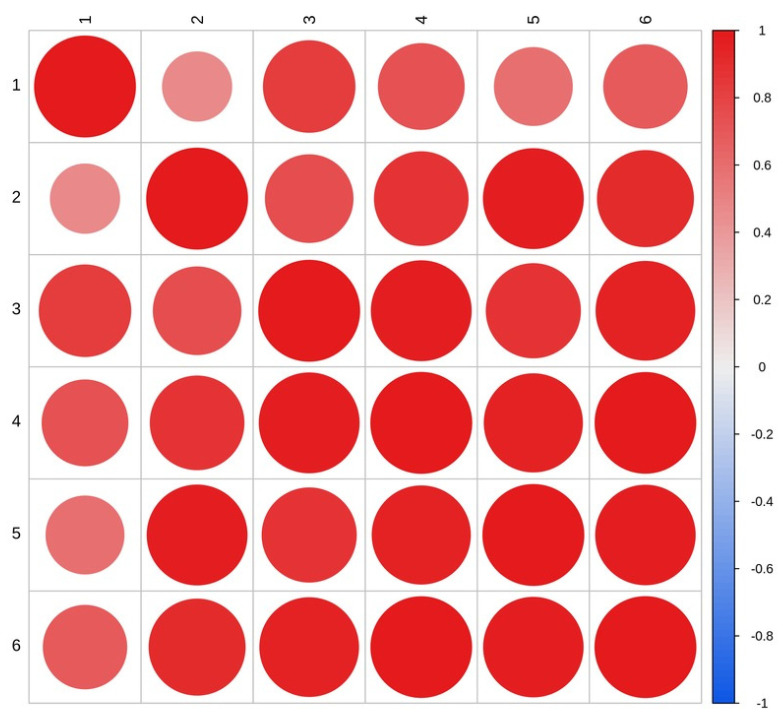
Correlation analysis in individual characteristics. (1) Gene expression levels under drought stress. (2) CAT activity under drought stress. (3) MDA activity under drought stress. (4) POD activity under drought stress. (5) PRO content under drought stress. (6) PRO activity under drought stress. The horizontal and vertical axes represent different samples, and the color difference in the pie chart represents a comparison of the Pearson correlation coefficient of gene expression in the two samples. The redder the color, the stronger the positive correlation, and the bluer the color, the weaker the correlation, while the larger the circle size, the stronger the correlation.

**Figure 8 ijms-25-09467-f008:**
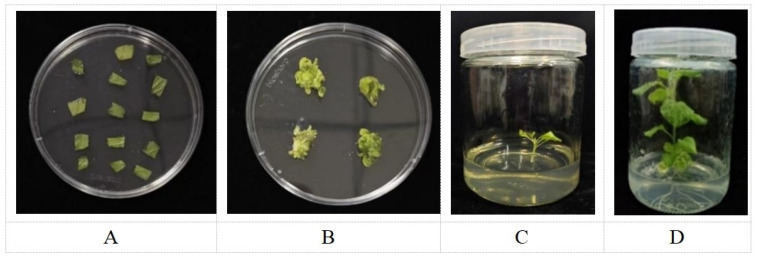
Flow chart of genetic transformation of AmGDSL1 transgenic tobacco. (**A**) Infected leaves. (**B**) Inducing callus. (**C**) Rooting culture. (**D**) Growing into a plant.

## Data Availability

Data are contained within the article.

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
