# Peer review of "Cloning and Functional Study of AmGDSL1 in Agropyron mongolicum"

_ijms, 2024, doi:10.3390/ijms25179467_

Round 1
Reviewer 1 Report
Comments and Suggestions for Authors
Dear Authors,
Reviewer comments ijms-3160501
The manuscript entitled „Cloning and functional study of AmGDSL1 in Agropyron mongolicum“ represents a useful study on the family of AmGDSL1 lipases in stress-hardy Agropyron mongolicum. The study deals with AmGDSL1 protein overexpression in Nicotiana benthamiana and studies the effects of osmotic stress (PEG-6000 treatment) on AmGDSL1-overexpressing plants regarding the levels of SOD, POD, CAT, MDA, and proline (PRO). The manuscript provides useful novel findings on the effects of AmGDSL1 overexpression on osmotic stress tolerance in transgenic tobacco plants; however, I have several important comments on the present manuscript which have to be addressed by the authors prior to the manuscript publication.
1/ In Materials and methods, the source of the plant material used including both Agropyron mongolicum (e.g., natural location – has to be specified) and Nicotiana benthamiana plants (lab name, institution) has to be given.
2/ Terminology: Use the term „osmotic stress“ instead of „drought stress“ for PEG-6000 treatment (PEG-6000 addition to 1/5 Hoagland nutrient solution). Use rather the term „dehydration stresses“ as a generalized term for „osmotic stress“, „drought“ and other related stresses based on a lack of available water for a plant.
3/ Results: In Figure 6 legend, ANOVA and Duncan´s multiple comparisons test have to be provided as statistical tools to determine significant differences indicated by lowercase letters above the data columns.
4/ Multidimensional statistics: From the data presented in Figure 6, it is obvious that some kind of multidimensional statistical analysis, e.g., principal component analysis, principal coordinate analysis, correlation analysis or cluster analysis, to provide a comprehensive view on the dynamics of the individual characteristics determined (SOD, POD, CAT, MDA, activity, proline content) in a broader context.
5/ Formal comments on the text:
Terminology: Figure 4 legend, line 190: Correct the typing error in the term: „electrophoreogram“ (not „electropherogram“).
Abbreviations: Although all abbreviations used in the text are explained when used for the first time, I think that a separate Abbreviations list should be included in the manuscript.
Taxonomy: Introduction, line 76: I do not know the term „wheat family in the Poaceae family.“ I think that it should rather be formulated as „Triticeae tribe in the Poaceae family“.
Introduction, line 52: Add a space between the word „responses“ and the following reference numbers.
Introduction, line 54: Add th word „which“ in the statement: „Its name is based on N-terminal which has a conserved GDSL motif….“
Introduction, line 75: Remove the word „which“ in the statement: „The GDSL genes as targets for genetic modification will be useful in improving crop yield, quality, or both.“
Results, line 118: Add a space between the words „time“ and „the seedlings“ in the statement: „At the same time, the seedlings of A. mongolicum…“
Line 123: Modify the word form as follows: „….the expression level of AmGDSL1 gene was significantly different.“
Results, line 162: Remove extra words „used to“ – twice in the same statement – in the statement: „The Agrobacterium liquid (LBA4404) containing the pCAMBIA1302-AmGDSL1 recombinant plasmid was used to transform the sterile young tobacco leaves…“
Results, line 288: Modify the word form „resulting“ to „resulted“
Materials and methods, lines 340-341: Modify the last statement in section 4.3. as follows: „The onion epidermal tissue was treated with 300 g/L sucrose solution for 5 minutes until the separation of the cell wall was observed.“
Conclusions, line 388: Use rather the term „dehydration stresses“ as a generalized term for „osmotic stress“, „drought“ and other related stresses based on a lack of available water for a plant, i.e., „AmGDSL1 overexpression lines showed significantly increased resistance to dehydration stresses.“
Final recommendation: reconsider after a major revision.
Comments on the Quality of English Language
Dear Authors,
Reviewer comments ijms-3160501
The manuscript entitled „Cloning and functional study of AmGDSL1 in Agropyron mongolicum“ represents a useful study on the family of AmGDSL1 lipases in stress-hardy Agropyron mongolicum. The study deals with AmGDSL1 protein overexpression in Nicotiana benthamiana and studies the effects of osmotic stress (PEG-6000 treatment) on AmGDSL1-overexpressing plants regarding the levels of SOD, POD, CAT, MDA, and proline (PRO). The manuscript provides useful novel findings on the effects of AmGDSL1 overexpression on osmotic stress tolerance in transgenic tobacco plants; however, I have several important comments on the present manuscript which have to be addressed by the authors prior to the manuscript publication.
1/ In Materials and methods, the source of the plant material used including both Agropyron mongolicum (e.g., natural location – has to be specified) and Nicotiana benthamiana plants (lab name, institution) has to be given.
2/ Terminology: Use the term „osmotic stress“ instead of „drought stress“ for PEG-6000 treatment (PEG-6000 addition to 1/5 Hoagland nutrient solution). Use rather the term „dehydration stresses“ as a generalized term for „osmotic stress“, „drought“ and other related stresses based on a lack of available water for a plant.
3/ Results: In Figure 6 legend, ANOVA and Duncan´s multiple comparisons test have to be provided as statistical tools to determine significant differences indicated by lowercase letters above the data columns.
4/ Multidimensional statistics: From the data presented in Figure 6, it is obvious that some kind of multidimensional statistical analysis, e.g., principal component analysis, principal coordinate analysis, correlation analysis or cluster analysis, to provide a comprehensive view on the dynamics of the individual characteristics determined (SOD, POD, CAT, MDA, activity, proline content) in a broader context.
5/ Formal comments on the text:
Terminology: Figure 4 legend, line 190: Correct the typing error in the term: „electrophoreogram“ (not „electropherogram“).
Abbreviations: Although all abbreviations used in the text are explained when used for the first time, I think that a separate Abbreviations list should be included in the manuscript.
Taxonomy: Introduction, line 76: I do not know the term „wheat family in the Poaceae family.“ I think that it should rather be formulated as „Triticeae tribe in the Poaceae family“.
Introduction, line 52: Add a space between the word „responses“ and the following reference numbers.
Introduction, line 54: Add th word „which“ in the statement: „Its name is based on N-terminal which has a conserved GDSL motif….“
Introduction, line 75: Remove the word „which“ in the statement: „The GDSL genes as targets for genetic modification will be useful in improving crop yield, quality, or both.“
Results, line 118: Add a space between the words „time“ and „the seedlings“ in the statement: „At the same time, the seedlings of A. mongolicum…“
Line 123: Modify the word form as follows: „….the expression level of AmGDSL1 gene was significantly different.“
Results, line 162: Remove extra words „used to“ – twice in the same statement – in the statement: „The Agrobacterium liquid (LBA4404) containing the pCAMBIA1302-AmGDSL1 recombinant plasmid was used to transform the sterile young tobacco leaves…“
Results, line 288: Modify the word form „resulting“ to „resulted“
Materials and methods, lines 340-341: Modify the last statement in section 4.3. as follows: „The onion epidermal tissue was treated with 300 g/L sucrose solution for 5 minutes until the separation of the cell wall was observed.“
Conclusions, line 388: Use rather the term „dehydration stresses“ as a generalized term for „osmotic stress“, „drought“ and other related stresses based on a lack of available water for a plant, i.e., „AmGDSL1 overexpression lines showed significantly increased resistance to dehydration stresses.“
Final recommendation: reconsider after a major revision.
Author Response
- Comment: In Materials and methods, the source of the plant material used including both Agropyron mongolicum (e.g., natural location – has to be specified) and Nicotiana benthamiana plants (lab name, institution) has to be given.
Response: Based on your suggestion, we added the details of the plant material in materials and methods. The seeds of A. mongolicum were collected in the germplasm resource garden of Inner Mongolia Agricultural University Grass Experiment Station. A. mongolicum seeds were selected and removed the palea and lemma, rinse with running water for 3 h, 75% ethanol for 30 s, rinse with sterilized distilled water 5 times, disinfect with 4.3% NaClO for 4 min, and rinse with sterilized distilled water 5 times. The seeds were placed in a germination box at 24℃ with 16 h/8h light and darkness in an artificial climate box (BIC-300). When seeds germinated and grew about 8-10 cm and be transplanted to a new germination box containing 1/5 Hoagland's nutrient solution (Lines 310-317).
The seeds of Nicotiana benthamiana are preserved in the laboratory (Agricultural College, Inner Mongolia Agricultural University). The seeds of Nicotiana benthamiana were vernalized at 4 ℃ for 2-3 days. They were then disinfected with 75% ethanol for 30 s, rinsed with sterile distilled water for 5 times, and 4.3% NaClO for 4 min. The seeds were placed in MS solid culture medium, and after germination, they were transferred to a new MS solid culture medium for genetic transformation experiments.The wild-type tobacco Nicotiana benthamiana (WT) and AmGDSL1-OE tobacco were cultured to consistent growth approximately 6 weeks of age, which subjected to drought treatment by cessation of watering for 7 days. (lines 325-331)
- 2. Comment:Terminology: Use the term „osmotic stress“ instead of „drought stress“ for PEG-6000 treatment (PEG-6000 addition to 1/5 Hoagland nutrient solution). Use rather the term „dehydration stresses“ as a generalized term for „osmotic stress“, „drought“ and other related stresses based on a lack of available water for a plant.
Response: Thank you for your suggestion, we checked the study and “drought stress” changed to “osmotic stress” for PEG-6000 treatment (Lines115-124 ).
- 3. Comment:Results: In Figure 6 legend, ANOVA and Duncan´s multiple comparisons test have to be provided as statistical tools to determine significant differences indicated by lowercase letters above the data columns.
Response: Based on your suggestion, ANOVA and Duncan´s multiple comparisons test have provided as statistical tools (Lines 402-405). In Figure 6, lower caseletters in the above graph represent significant differences at the 0.05 level. At the same time, significant differences are marked in Figure 1 and Figure 5.
- Comment:Multidimensional statistics: From the data presented in Figure 6, it is obvious that some kind of multidimensional statistical analysis, e.g., principal component analysis, principal coordinate analysis, correlation analysis or cluster analysis, to provide a comprehensive view on the dynamics of the individual characteristics determined (SOD, POD, CAT, MDA, activity, proline content) in a broader context.
Response: Based on your suggestion, we conducted correlation analysis individual characteristics determined (SOD, POD, CAT, MDA activity, proline content) in a broader context. The results showed that under drought stress, The SOD, POD, CAT, MDA activity and proline content were positively correlated with gene expression (Figure.7)(Lines 227-229).
Figure 7. Correlation analysis in individual characteristics. 1: Gene expression levels under drought stress. 2: The CAT activity under drought stress. 3: The MDA activity under drought stress. 4: The POD activity under drought stress. 5: The PRO content under drought stress. 6: The PRO activity under drought stress.
- Comment:Terminology: Figure 4 legend, line 190: Correct the typing error in the term: „electrophoreogram“ (not „electropherogram“).
Response: Based on your suggestion, Figure 4 legend has been revised. And in Figure 2 legend has been revised (Lines 156, Lines 189).
- Comment:Abbreviations: Although all abbreviations used in the text are explained when used for the first time, I think that a separate Abbreviations list should be included in the manuscript.
Response: Based on your suggestion,We have added Abbreviations list in the manuscript.
|
Abbreviation |
Full name |
|
2,4-D |
2,4-Dichlorophenoxyacetic acid |
|
6-BA |
6-Benzylaminopurine |
|
CAT |
Catalase |
|
IAA |
Indole acetic acid |
|
Kana |
Kanamycin |
|
MDA |
Malondialdehyde |
|
NAA |
1-Naphthaleneacetic acid |
|
PCR |
Polymerase Chain Reaction |
|
POD |
Peroxidase |
|
PRO |
Proline |
|
qRT-PCR |
Real time quantitative PCR |
|
Rif |
Rifampicin |
|
SOD |
Superoxide Dismutase |
|
TMT |
Timentin |
|
WT |
wild-type tobacco |
- Comment:Taxonomy: Introduction, line 76: I do not know the term „wheat family in the Poaceae family.“ I think that it should rather be formulated as „Triticeae tribe in the Poaceae family“.
Response: Based on your suggestion,We have made modifications (Lines 76).
- Comment:Introduction, line 52: Add a space between the word „responses“ and the following reference numbers.
Response: Based on your suggestion,We have made modifications (Lines 52).
- Comment:Introduction, line 54: Add the word „which“ in the statement: „Its name is based on N-terminal which has a conserved GDSL motif….“
Response: Based on your suggestion,We have made modifications.(Lines 53).
- Comment:Introduction, line 75: Remove the word „which“ in the statement: „The GDSL genes as targets for genetic modification will be useful in improving crop yield, quality, or both.“
Response: Based on your suggestion,We have made modifications.(Lines 72-74).
- Comment:Results, line 118: Add a space between the words „time“ and „the seedlings“ in the statement: „At the same time, the seedlings of A. mongolicum…“
Response: Based on your suggestion,We have made modifications.(Lines 115).
12. Comment: Line 123: Modify the word form as follows: „….the expression level of AmGDSL1 gene was significantly different.“
Response: Based on your suggestion,We have made modifications.(Lines 116-117)
- Comment:Results, line 162: Remove extra words „used to“ – twice in the same statement – in the statement: „The Agrobacterium liquid (LBA4404) containing the pCAMBIA1302-AmGDSL1 recombinant plasmid was used to transform the sterile young tobacco leaves…“
Response: Based on your suggestion,We have made modifications.(Lines 159-160)
- Comment:Results, line 288: Modify the word form „resulting“ to „resulted“
Response: Based on your suggestion,We have made modifications.(Lines 295)
- Comment:Materials and methods, lines 340-341: Modify the last statement in section 4.3. as follows: „The onion epidermal tissue was treated with 300 g/L sucrose solution for 5 minutes until the separation of the cell wall was observed.“
Response: Based on your suggestion,We have made modifications(Lines 363-364).
- Comment:Conclusions, line 388: Use rather the term „dehydration stresses“ as a generalized term for „osmotic stress“, „drought“ and other related stresses based on a lack of available water for a plant, i.e., „AmGDSL1 overexpression lines showed significantly increased resistance to dehydration stresses.“
Response: Based on your suggestion,We have made modifications.(Lines 409-415)

Reviewer 2 Report
Comments and Suggestions for Authors
The paper performs a complete study of a gene involved in drought resistance. The study is complete and the amount of novel information meets with the requirements for being accepted in IJMS. But some figures need improvement and better controls in order to strengthen the conclusions.
Figure 2C: enhance the quality, as is difficult to read. Also there are chinese characters which are difficult to understand for general readers.
Figure 3: The quality is very poor. Please, provide images with better resolution. In addition the problem is that the control is GFP alone. If the fussion properly in unproperly processed and is cut, teh results is going to be the same. Can you provide a western blot from transiently expressed tissue to check the integrity of the fussion protein?
Figure 4: Please, include a control of the RT PCR of a wild type tobacco and the CK transformed with the empty plasmid, as endogenous tobacco orthologues may interfere with the analysis.
Author Response
- Comment:Figure 2C: enhance the quality, as is difficult to read. Also there are chinese characters which are difficult to understand for general readers.
Response: Based on your suggestion,We have made modifications.
Figure 2C. Alignment of AmGDSL1 transcribed sequence with sequencing sequence.
- Comment:Figure 3: The quality is very poor. Please, provide images with better resolution. In addition the problem is that the control is GFP alone. If the fussion properly in unproperly processed and is cut, teh results is going to be the same. Can you provide a western blot from transiently expressed tissue to check the integrity of the fussion protein?
Response: Based on your suggestion, The Figure 3 we have made revision. Due to laboratory conditions, western blot could not be performed. For the accuracy of the experiment, we conducted an empty control in subcellular localization.
Figure 3. Subcellular localization of AmGDSL1. A: EGFP positive control (pCAMBIA1302-EGFP). B: pCAMBIA1302-AmGDSL1 EGFP is independently expressed in onion epidermal cells. C: Expression of pCAMBIA1302-AmGDSL1-EGFP after plasma wall separation. Scale bar=50um.
- Comment:Figure 4: Please, include a control of the RT PCR of a wild type tobacco and the CK transformed with the empty plasmid, as endogenous tobacco orthologues may interfere with the analysis.
Response: Based on your suggestion, the CK transformed with the empty plasmid for control of the qRT-PCR.
Figure.4 Relative expression levels of the AmGDSL1 gene in transgenic plants. Different lowercase letters indicate significant differences at the 0.05 level.

Reviewer 3 Report
Comments and Suggestions for Authors
The topic of this study is significant and interesting, focusing on the identification of the AmGDSL1 gene in A. mongolicum under drought conditions and its potential role in drought stress responses. The manuscript is generally well-written, the experiments are well-executed, and the results are clearly presented. However, before publication, the following minor issues need to be addressed:
Comment 1: Line 13: GDSL lipase or esterases/lipase
Comment 2: Line 28: please add references.
Comment 3: Line 29 : Are you sure about the accuracy of this statement? The reference provided does not seem to correspond to the sentence.
Comment 4: Figure 2. need to improve quality
Comment 5: Line 250 and 264: add the number of references.
Comment 6: Line 302: please give the used accession name of A. mongolicum
Comment 7: Line 303: On what basis did the authors choose a concentration of 0.042 mol/L PEG-6000?
Comment 8: Line 306: please give more details about the growth place and conditions.
Comment 9: Line 343: The description of how the AmGDSL1 gene was introduced could be more detailed, including any specifics about the plasmid preparation or Agrobacterium strain used.
Comment 10: Line 384: The conclusion is brief , and the future perspectives are missing in the conclusion section.
Comments on the Quality of English LanguageMinor editing of English language required.
Author Response
- Comment: Line 13: GDSL lipase or esterases/lipase
Response: Based on your suggestion,We have made modifications.(Line 13)
- Comment: Line 28: please add references.
Response: Based on your suggestion,We have added references.
1 Yang, P., Zhai, X., Huang, H., Zhang, Y.; Zhu, Y., Shi, X., Zhou, L., Fu, C. Association and driving factors of meteorological drought and agricultural drought in Ningxia, Northwest China. Atmospheric Research. 2023, 289, 106753.
- Comment: Line 29 : Are you sure about the accuracy of this statement? The reference provided does not seem to correspond to the sentence.
Response: Thank you for your suggestion, this statement was inappropriate,we are deleted it.
- Comment : Figure 2. need to improve quality
Response: Based on your suggestion,We have made modifications.
Figure 2. Cloning of AmGDSL1 gene and vector construction. A: PCR amplification electrophoresis detection of AmGDSL1. M:D2000 DNA Maker, 1-6: AmGDSL1. B: PCAMBIA1302-AmGDSL1 transformed Agrobacterium LBA4404 colony PCR electrophoreogram. C: Alignment of AmGDSL1 transcribed sequence with sequencing sequence.
- Comment : Line 250 and 264: add the number of references.
Response: Based on your suggestion,We are added the references.
- Comment: Line 302: please give the used accession name of A. mongolicum
Response: Based on your suggestion,We have submitted the sequence to NCBI and the submission number 2861337, but the accession number has not been obtained.
- Comment:Line 303: On what basis did the authors choose a concentration of 0.042 mol/L PEG-6000?
Response: 0.042 mol/L PEG-6000 is 25% PEG-6000. According previous study, 20% PEG-6000 dehydration stresses was optimal [1-3]. The A. mongolicum treated with 20% PEG-6000 , there was no significant wilting of the leaves. When A. mongolicum treated with 25% PEG-6000, the seedlings decreased and was inhibited. Compared with 30% PEG-6000 treatment, 25% PEG-6000 was selected to simulate drought stress in order to better observe the mechanism of drought stress in A. mongolicum wilting [4].
1 Lu, A., Su, L.T., Fan, N., Wen, W., Gao, L. The MsDHN1-MsPIP2 1-MsmMYB module orchestrates the trade-off between growth and survival of alfalfa in response to drought stress. Plant Biotechnol J. 2023, 1, 1-14.
2 Zhang, T., Zhang, W., Ding, C., Hu, Z., Fan, C., Zhang, J., Li, Z. A breeding strategy for improving drought and salt tolerance of poplar based on CRISPR/Cas9. Plant Biotechnol J. 2023, 21, 2160-2162.
3 Wei, W., Lu, L., Bian, X.H., Li, Q.T., Han,J.Q., Tao,J.J. Zinc-finger protein GmZF351 improves both salt and drought stress tolerance in soybean. J Integr Plant Biol. 2023, 65, 1636-1650.
4 Zhang, X.F., Ma, Y.H., Fan, B.B., Sun, F.C., Zhai, Y.Q., Zhao, Y. The Agropyron mongolicum bHLH Gene AmbHLH148 Positively Involved in Transgenic Nicotiana benthamiana Adaptive Response to Drought Stress. Agronomy. 2023, 13, 2918.
- Comment: Line 306: please give more details about the growth place and conditions.
Response: Based on your suggestion, we added the details of the plant material in materials and methods. The seeds of A. mongolicum were collected in the germplasm resource garden of Inner Mongolia Agricultural University Grass Experiment Station. A. mongolicum seeds were selected and removed the palea and lemma, rinse with running water for 3 h, 75% ethanol for 30 s, rinse with sterilized distilled water 5 times, disinfect with 4.3% NaOCl for 4 min, and rinse with sterilized distilled water 5 times. The seeds were placed in a germination box at 24℃ with 16 h/8h light and darkness in an artificial climate box (BIC-300). When seeds germinated and grew about 8-10 cm and be transplanted to a new germination box containing 1/5 Hoagland's nutrient solution (Lines 310-317).
The seeds of Nicotiana benthamiana are preserved in the laboratory (Agricultural College, Inner Mongolia Agricultural University). The seeds of Nicotiana benthamiana were vernalized at 4 ℃ for 2-3 days. They were then disinfected with 75% ethanol for 30 s, rinsed with sterile distilled water for 5 times, and 4.3% NaClO for 4 min. The seeds were placed in MS solid culture medium, and after germination, they were transferred to a new MS solid culture medium for genetic transformation experiments.The wild-type tobacco Nicotiana benthamiana (WT) and AmGDSL1-OE tobacco were cultured to consistent growth approximately 6 weeks of age, which subjected to drought treatment by cessation of watering for 7 days. (lines 325-331)
- Comment:Line 343: The description of how the AmGDSL1 gene was introduced could be more detailed, including any specifics about the plasmid preparation or Agrobacterium strain used.
Response:pCAMBIA1302-EGFP plasmids (Abiowell, Changsha, China) and pEASY-AmGDSL1 plasmids and extracted according to the plasmid extraction kit (TIANGEN DP103, Beijing, China) and was digested using NcoI and SpeI restriction endonucleases (Takara, Beijing, China).The pCAMBIA1302-AmGDSL1-EGFP recombinant vector was constructed by ligating the AmGDSL1 product without a termination codon into the pCAMBIA1302-EGFP vector fragment. The enzyme digestion system included 0.5 mL NcoI, 0.5 mL SpeI, 2 mL 2×K Buffer, 10 mL pCAMBIA1302-EGFP/pEASY-AmGDSL1 and 7 mL ddH2O. The digestion process was carried out at 37℃ for 16 h. The enzyme digestion products was used DNA Gel/PCR Purification Miniprep Kit (DC3511, Hangzhou, China). The ligation system included 2.5 µL of linearized pCAMBIA1302 vector, 1.5 µL AmGDSL1, 2 µL T4 DNA buffer, 1 µL DNA Ligation Kit (TaKaRa, Code No. 6022Q, Beijing, China), and 13 µL ddH2O. The digestion process was carried out overnight at 16℃. The ligation products took place on the ice for 25 min, water bath at 42℃ for 45 s and the ice for 2 min that transformed the ligation products into E.coli receptor TOP10 cells. Positivetrans formants were selected on LB agar plates containing 50 µg/mL kan for PCR analysis. After the PCR detection (AmGDSL1-F/R) of the recombinant plasmid, it was named pCAMBIA1302-AmGDSL1. The pCAMBIA1302-AmGDSL1 plasmid liquid nitrogen for 5 min, water bath at 37℃ for 5 min and the ice for 5 min transformed into agrobacterium LBA4404 competent cells (Weidi, Shanghai, China). Positivetrans formants were selected on LB agar plates containing 50 µg/mL kanamycin for PCR analysis. (Lines 343-363).
- Comment: Line 384: The conclusion is brief , and the future perspectives are missing in the conclusion section.
Response: Based on your suggestion, the conclusion have made revision.
In this study, a typical GDSL gene AmGDSL1 was cloned from A. mongolicum and functionally verified in overexpressing tobacco. AmGDSL1 is a cytoplasm-localized protein. We also examined the increase in AmGDSL1 gene expression induced by PEG6000 treatment of A. mongolicum. AmGDSL1 overexpression lines showed significantly increased resistance to dehydration stresses. This study explores the GDSL gene related to dehydration stresses in A. mongolicum, providing reference for crop genetic improvement. Future research may improve crop resistance in gene breeding. (Lines 420-430).

Round 2
Reviewer 1 Report
Comments and Suggestions for Authors
Dear Authors,
Reviewer comments ijms-3160501v2
The revised manuscript entitled „Cloning and functional study of AmGDSL1 in Agropyron mongolicum“ was significantly improved by the authors. However, I still have some comments on the revised manuscript which are provided below:
1/ Terminology: It should be clear from the manuscript that PEG-6000 treatment is an osmotic stress which is only used to simulate drought while drought stress means a lack of water in soil or solid substrate.
2/ In Figure 1, 4,5,6 legends, the kind of statistical test used to determine significant differences between the samples at 0.05 level as indicated by different letters above the data columns has to be given.
3/ In Figure 7 legend, not only the different colour of the circles corresponding to the different values of statistical coefficient but also the different sizes of the circles in the correlation matrix have to be explained. What kind of correlation coefficient is determined in the correlation matrix, Pearson´s or Spearman´s coefficient??
4/ In Figure 3, the cale bars are clearly visible only in part A while in parts B and C, the scale bars visibility has to be improved.
5/ In Materials and methods, some information on just ANOVA and DMRT 0.05 test are provided at the end of part 4.6.; however, I think that special part 4.7. Statistical analysis including not only basic information on ANOVA and DMRT test but also on correlation analysis provided in Figure 7 has to be added.
6/ In Discussion, in addition to GDSL genes, GELP genes are also discussed without any brief explanation. I think that some brief explanation or justification has to be provided by the authors to ensure better orientation of the readers in the text.
7/ Formal comments on the text:
In Abbreviations list, I think that also the gene names such as GDSL and GELP have to be added.
Introduction, line 59: Remove extra „in“ in the stetament „…may be involved in regulating the drought stress in pigeon pea….“
Introduction, line 70: Correct the verb form „can improve“ (not „can improved“) in the statement: „Overexpression of CpGLIP1 can improve tolerance to drought and cold…“
Introduction, line 77: Write „An abundance of drought stress responsive genes…“ or „Several drought stress responsive gens…“
Introduction, line 88: Add a comma following the words: „Under drought conditions,…“
Results, line 116: Add a verb „revealed“ in the statement: „Notably, as the osmotic stress duration increases, the expression level of AmGDSL1 gene revealed an overal increasing trend.“
Line 215: Modify the word form „regulates“ to „regulation“ in the heading: „2.7. Expression regulation in AmGDSL1 transgenic tobacco“.
Discussion, line 259: Remove the word „in“ in the statetemnt: „Our study shows that the AmGDSL1 gene has the highest expression level…“
Discussion, line 298: Modify the verb form „identificated“ to „identified“ and remove extra word „but“ in the statemenmt: „In our study, the drought tolerance function of the AmGDSL1 gene was preliminary identified but its metabolic pathways and molecular mechanisms involved in dehydration stress response needed further research.“
Materials and methods, line 347: Correct the space position in the words „Positive transformants…“
Materials and methods, line 372: Write the palnt name „Nicotiana benthamiana“ in italics.
Final recommendation: Accept after a minor revision.
Comments on the Quality of English LanguageDear Authors,
Reviewer comments ijms-3160501v2
The revised manuscript entitled „Cloning and functional study of AmGDSL1 in Agropyron mongolicum“ was significantly improved by the authors. However, I still have some comments on the revised manuscript which are provided below:
1/ Terminology: It should be clear from the manuscript that PEG-6000 treatment is an osmotic stress which is only used to simulate drought while drought stress means a lack of water in soil or solid substrate.
2/ In Figure 1, 4,5,6 legends, the kind of statistical test used to determine significant differences between the samples at 0.05 level as indicated by different letters above the data columns has to be given.
3/ In Figure 7 legend, not only the different colour of the circles corresponding to the different values of statistical coefficient but also the different sizes of the circles in the correlation matrix have to be explained. What kind of correlation coefficient is determined in the correlation matrix, Pearson´s or Spearman´s coefficient??
4/ In Figure 3, the cale bars are clearly visible only in part A while in parts B and C, the scale bars visibility has to be improved.
5/ In Materials and methods, some information on just ANOVA and DMRT 0.05 test are provided at the end of part 4.6.; however, I think that special part 4.7. Statistical analysis including not only basic information on ANOVA and DMRT test but also on correlation analysis provided in Figure 7 has to be added.
6/ In Discussion, in addition to GDSL genes, GELP genes are also discussed without any brief explanation. I think that some brief explanation or justification has to be provided by the authors to ensure better orientation of the readers in the text.
7/ Formal comments on the text:
In Abbreviations list, I think that also the gene names such as GDSL and GELP have to be added.
Introduction, line 59: Remove extra „in“ in the stetament „…may be involved in regulating the drought stress in pigeon pea….“
Introduction, line 70: Correct the verb form „can improve“ (not „can improved“) in the statement: „Overexpression of CpGLIP1 can improve tolerance to drought and cold…“
Introduction, line 77: Write „An abundance of drought stress responsive genes…“ or „Several drought stress responsive gens…“
Introduction, line 88: Add a comma following the words: „Under drought conditions,…“
Results, line 116: Add a verb „revealed“ in the statement: „Notably, as the osmotic stress duration increases, the expression level of AmGDSL1 gene revealed an overal increasing trend.“
Line 215: Modify the word form „regulates“ to „regulation“ in the heading: „2.7. Expression regulation in AmGDSL1 transgenic tobacco“.
Discussion, line 259: Remove the word „in“ in the statetemnt: „Our study shows that the AmGDSL1 gene has the highest expression level…“
Discussion, line 298: Modify the verb form „identificated“ to „identified“ and remove extra word „but“ in the statemenmt: „In our study, the drought tolerance function of the AmGDSL1 gene was preliminary identified but its metabolic pathways and molecular mechanisms involved in dehydration stress response needed further research.“
Materials and methods, line 347: Correct the space position in the words „Positive transformants…“
Materials and methods, line 372: Write the palnt name „Nicotiana benthamiana“ in italics.
Final recommendation: Accept after a minor revision.
Author Response
Reviewer #1:
The revised manuscript entitled „Cloning and functional study of AmGDSL1 in Agropyron mongolicum“ was significantly improved by the authors. However, I still have some comments on the revised manuscript which are provided below:
- Comment: Terminology: It should be clear from the manuscript that PEG-6000 treatment is an osmotic stress which is only used to simulate drought while drought stress means a lack of water in soil or solid substrate.
Response: Thank you for your suggestion, we checked the study and “drought stress” changed to “osmotic stress” for PEG-6000 treatment (Line317, Lines 402-405 ). AmGDSL1-OE tobacco were subjected to drought treatment by cessation of watering for 7 days, we use ‘drought treatment’ in this section.
- Comment: In Figure 1, 4,5,6 legends, the kind of statistical test used to determine significant differences between the samples at 0.05 level as indicated by different letters above the data columns has to be given.
Response: The Duncan's test used to determine significant differences between the samples at 0.05 level as indicated by different letters above the data columns had be given (In Figure 1, 4,5,6 legends).
- Comment:In Figure 7 legend, not only the different colour of the circles corresponding to the different values of statistical coefficient but also the different sizes of the circles in the correlation matrix have to be explained. What kind of correlation coefficient is determined in the correlation matrix, Pearson´s or Spearman´s coefficient?
Response: The horizontal and vertical axes represent different samples, and the color difference in the pie chart represents the Pearson correlation coefficient of gene expression between two samples. The redder the color, the stronger the positive correlation, and the bluer the color, the weaker the correlation. And the larger the circle size, the stronger the correlation (Lines 243-247 ).
- Comment:In Figure 3, the cale bars are clearly visible only in part A while in parts B and C, the scale bars visibility has to be improved.
Response: Based on your suggestion,We have made modifications.
- Comment:In Materials and methods, some information on just ANOVA and DMRT 0.05 test are provided at the end of part 4.6.; however, I think that special part 4.7. Statistical analysis including not only basic information on ANOVA and DMRT test but also on correlation analysis provided in Figure 7 has to be added.
Response: Based on your suggestion, We have added data statistics.
4.7. Statistical analysis
Data were collated and bar charts were drawn using Excel 2010 (Microsoft, USA). Data were statistically analyzed by one-way variance analysis by Duncan’s multiple range tests in SPSS 19.0 (IBM Inc., USA) software. And differences were regarded as significant at p <0.05. The Pearson correlation coefficients (r) were computed using R software (psych 2.2.9). (Lines 423-425).
- Comment:In Discussion, in addition to GDSL genes, GELP genes are also discussed without any brief explanation. I think that some brief explanation or justification has to be provided by the authors to ensure better orientation of the readers in the text.
Response: Based on your suggestion, we have made revisions to this term. The GELP is the abbreviation for GDSL-type Esterase/Lipase (Lines 52).
- Comment:In Abbreviations list, I think that also the gene names such as GDSL and GELP have to be added.
Response: Based on your suggestion, we have added it.
- Comment:Introduction, line 59: Remove extra „in“ in the stetament „…may be involved in regulating the drought stress in pigeon pea….“
Response: Based on your suggestion,We have made modifications.
- Comment:Introduction, line 70: Correct the verb form „can improve“ (not „can improved“) in the statement: „Overexpression of CpGLIP1 can improve tolerance to drought and cold…“
Response: Based on your suggestion,We have made modifications.
- Comment:Introduction, line 77: Write „An abundance of drought stress responsive genes…“ or „Several drought stress responsive gens…“
Response: Based on your suggestion,We have made modifications.
- Comment:Introduction, line 88: Add a comma following the words: „Under drought conditions,…“
Response: Based on your suggestion,We have made modifications.
- Comment:Results, line 116: Add a verb „revealed“ in the statement: „Notably, as the osmotic stress duration increases, the expression level of AmGDSL1 gene revealed an overal increasing trend.“
Response: Based on your suggestion,We have made modifications.
- Comment:Line 215: Modify the word form „regulates“ to „regulation“ in the heading: „2.7. Expression regulation in AmGDSL1 transgenic tobacco“.
Response: Based on your suggestion,We have made modifications.
- Comment:Discussion, line 259: Remove the word „in“ in the statetemnt: „Our study shows that the AmGDSL1 gene has the highest expression level…“
Response: Based on your suggestion,We have made modifications.
- Discussion, line 298: Modify the verb form „identificated“ to „identified“ and remove extra word „but“ in the statemenmt: „In our study, the drought tolerance function of the AmGDSL1 gene was preliminary identified but its metabolic pathways and molecular mechanisms involved in dehydration stress response needed further research.“
Response: Based on your suggestion,We have made modifications.
- Materials and methods, line 347: Correct the space position in the words „Positive transformants…“
Response: Based on your suggestion,We have made modifications.
- Materials and methods, line 372: Write the palnt name „Nicotiana benthamiana“ in italics.
Response: Based on your suggestion,We have made modifications.

Reviewer 2 Report
Comments and Suggestions for Authors
Paper has been improved. I can recommend publication.
Author Response
Thank you for your suggestion.